# ON THE INTERPOLATION EFFECT OF SCORE SMOOTHING

## ABSTRACT

Score-based diffusion models have achieved remarkable progress in various domains with an ability to generate new data samples that do not exist in the training set. In this paper, we examine a hypothesis that this phenomenon manifests an interpolation effect caused by a smoothing of the empirical score function. Focusing on settings where the training set lies in a one-dimensional linear subspace, we take a distribution-agnostic perspective and study the interplay between score smoothing and the denoising dynamics with mathematically solvable models. We demonstrate how score smoothing can lead to the generation of samples that interpolate among the training data within the subspace while avoiding a full memorization of the training set.

## 1 INTRODUCTION

In the past years, score-based diffusion models have emerged as an important pillar of generative modeling in a variety of domains from image and video generation to drug discovery (Sohl-Dickstein et al., 2015; Song & Ermon, 2019; Ho et al., 2020; Ramesh et al., 2022; Brooks et al., 2024). For instance, after being trained on datasets of actual images or existing molecular configurations, such models are capable of transform noise samples into high-quality images or chemically-plausible molecules. Importantly, the generated samples do not belong to the original training set, indicating an exciting potential of such models to generalize beyond the training data and, in a sense, be creative.

The creativity of score-based diffusion models has not been fully understood from a theoretical point of view. At the core of these models is the training of neural networks (NNs) to fit a series of target functions – called the *empirical score functions (ESF)* – which will be used to drive the denoising process. The precise form of these functions are determined by the training set and can be computed exactly in principle (though inefficient in practice). However, when equipped with the exact precise ESF instead of the approximate version learned by NNs, the model will end up generate data points that already exist in the training set (Yi et al., 2023; Li et al., 2024), a phenomenon often called *memorization*. This suggests that, for the models to generalize fresh samples beyond the training set, it is crucial to have certain regularizations on the score function (perhaps implicitly through NN training) that prevent the ESF to be learned exactly. In particular, Scarvelis et al. (2023) hypothesized that smoothing the ESF results in a tendency of the generated samples to *interpolate* among training data points, but it remains unclear mathematically *how* score smoothing leads to interpolation.

Meanwhile, a different yet related phenomenon of generative models is *hallucination*, in which they generate samples that are qualitatively (and often undesirably) different from the training set. Fundamentally, both generalization and hallucination concern scenarios where models generate samples that are distinct from the original training set, with their difference being whether the the new samples lie within or out of the support of the underlying target distribution. Hence, the hallucination phenomenon is also related to the tendency of diffusion models to interpolate among or extrapolate from the training set. Indeed, the recent work of Aithal et al. (2024) proposed "mode interpolation" as the essence of hallucination and demonstrates its empirical presence when the NN learns a smoother version of the ESF, but its mathematical mechanism remains unclear as well.

In this work, we will illustrate mathematically how score smoothing in diffusion models could result in data interpolation and subspace recovery. Focusing on examples where the training data belong to a one-dimensional subspace, we consider a simple type of local smoothing on the ESF and derive

that it can lead the denoising dynamics to recover a non-singular density that is supported on the one-dimensional subspace, thus recovering the subspace from training data.

## 1.1 NOTATIONS

If $p$ is a smooth probability density on $\mathbb{R}^d$, we call $\nabla \log p : \mathbb{R}^d \to \mathbb{R}^d$ its *score function* (Hyvärinen & Dayan, 2005). We let $p_{\mathcal{N}}(x; \sigma) = (\sqrt{2\pi}\sigma)^{-1} \exp(-x^2/(2\sigma^2))$ denote the 1-D Gaussian density with mean 0 and variance $\sigma^2$. We write $\text{sgn}(x)$ for the sign of $x \in \mathbb{R}$ and $[n] := \{1, .., n\}$ for $n \in \mathbb{N}_+$.

## 2 BACKGROUND

While score-based diffusion models have many variants, we will focus on a basic one (called the Variance Exploding version by Song et al. 2021b) for simplicity, where the *forward (or noising) process* is defined by the following stochastic differential equation (SDE) in $\mathbb{R}^d$ for $t \geq 0$:

$$d\mathbf{x}_t = d\mathbf{w}_t , \quad \mathbf{x}_0 \sim p_0 , \tag{1}$$

where $\mathbf{w}$ is the Wiener process (a.k.a. Brownian motion) in $\mathbb{R}^d$. The marginal distribution of $\mathbf{x}_t$, denoted by $p_t$, is thus fully characterized by the initial one $p_0$ together with the conditional distributions, $p_{t|0}(\boldsymbol{x}|\boldsymbol{x}') = \prod_{i=1}^d p_{\mathcal{N}}(x_i - x_i'; \sqrt{t})$. A key observation is that this process is equivalent in distribution to a *deterministic* dynamics often called the *probability flow ODE* (Song et al., 2021b) that is driven by the family of score functions of $p_t$,

$$d\mathbf{x}_t = -\tfrac{1}{2}\boldsymbol{s}_t(\mathbf{x}_t)dt , \tag{2}$$

$$\text{where} \quad \boldsymbol{s}_t(\mathbf{x}) = \nabla \log p_t(\mathbf{x}) . \tag{3}$$

In generative modeling, $p_0$ is often a distribution of interest that is hard to sample directly (e.g. the distribution of cat images in pixel space), while when $T$ is large, $p_T$ is always close to a Gaussian distribution (with variance increasing in $T$), from which samples are easy to obtain. Thus, to obtain samples from $p_0$, a insightful idea is to first sample from $p_T$ and then follow the *reverse (or denoising) process* by simulating (2) backward-in-time.

A main challenge in such a procedure lies in the estimation of the family of score functions $\nabla \log p_t$ for $t \in [0, T]$. In reality, we have no prior knowledge of each $p_t$ (or even $p_0$) but only a training set $S = \{\boldsymbol{y}_k\}_{k \in [n]}$ that is assumed to be sampled from $p_0$. This allows us to define an *empirical* version of the noising process where the SDE (1) is now initialized with the marginal distribution at $t = 0$ being uniform over the set $S$ (i.e., $p_0^{(n)} = \frac{1}{n}\sum_{k=1}^n \delta_{\boldsymbol{y}_k}$), and thus $\mathbf{x}_t$ is distributed as $p_t^{(n)}(\boldsymbol{x}) := \frac{1}{n}\sum_{k=1}^n p_{t|0}(\boldsymbol{x}|\boldsymbol{y}_k)$. In practice, one often uses a neural network (NN) to parameterize a function $s_{\boldsymbol{\theta}}(x, t)$ that approximates the *empirical score function (ESF)*, $\nabla \log p_t^{(n)}$, and it will serve as a proxy for $\nabla \log p_t$ in (2) to drive the denoising process. The NN parameters are usually trained to minimize variants of the following loss function, which is a squared loss averaged over $p_t^{(n)}$ (we will call it the *empirical $L^2$ discrepancy*) integrated over time (Song et al., 2021b):

$$\min_{\boldsymbol{\theta}} \int_0^T t \mathbb{E}_{\mathbf{x} \sim p_t^{(n)}}[\|s_{\boldsymbol{\theta}}(\mathbf{x}, t) - \nabla \log p_t^{(n)}(\mathbf{x})\|^2]dt . \tag{4}$$

In practice, the minimization problem (4) is often solved numerically via Monte-Carlo sampling combined with ideas from Hyvärinen & Dayan (2005); Vincent (2011), but we know that the minimum is attained uniquely by the ESF itself, which can be computed in closed form based on $S$ (details to be given below). So what if we plug the ESF directly into the denoising dynamics (2) instead of an NN approximation? In that case, we end up with an empirical version of the denoising process:

$$d\mathbf{x}_t = -\tfrac{1}{2}\nabla \log p_t^{(n)}(\mathbf{x}_t) , \tag{5}$$

and the outcome at $t = 0$ will inevitably be $p_0^{(n)}$. In other words, the model *memorizes* the training data instead of generating fresh samples. This suggests that the creativity of the diffusion model hinges on a *sub-optimal* solution to (4) and an *imperfect* approximation to the ESF. Indeed, the memorization phenomenon can be observed in practice when the models have large capacities relative to the training set size (Gu et al., 2023; Kadkhodaie et al., 2024), which likely results in too good an

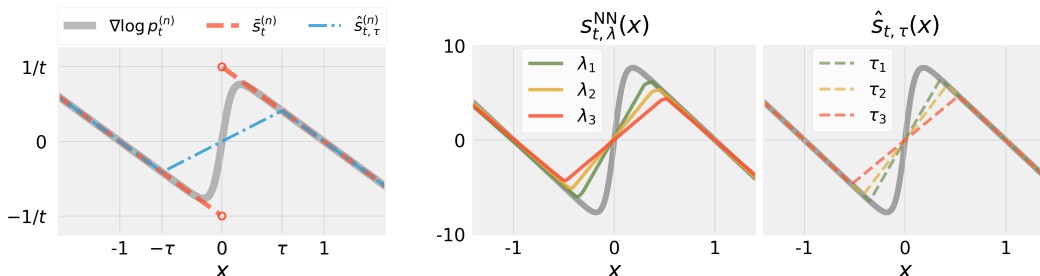

Figure 1: Illustrating the empirical score function (ESF) and its substitutes in the case of two training data points in one dimension discussed in Section 3. **Left**: Comparing the ESF ($\nabla \log p_t^{(n)}$), the piece-wise linear approximation ($\bar{s}_t^{(n)}$), and the $\tau$-smoothed version of the latter ($\hat{s}_{t,\tau}^{(n)}$). **Center**: NN-learned score functions ($s_{t,\lambda}^{\text{NN}}$) under increasing strengths of regularization, $\lambda$. **Right**: $\hat{s}_{t,\tau}^{(n)}$ under increasing scales of smoothing, $\tau$. Additional details are given in Appendix A.7.

approximation to the ESF. This leads to the hypothesis that regularizing the score estimator gives rise to the model's ability to generalize out of the training set, though a theoretical understanding of the mechanism is still under development. In this work, we will focus on simple setups to show mathematically how smoothing the ESF can enable the generation of new samples that interpolate among the training set.

## 3 MATHEMATICAL ANALYSIS

We start with a simplest setup where $d = 1$ and $S = \{-1, 1\}$ consists of only two points. In this case, at time $t$, the noised empirical distribution is $p_t^{(n)}(x) = \frac{1}{2}(p_{\mathcal{N}}(x+1; \sqrt{t}) + p_{\mathcal{N}}(x-1; \sqrt{t}))$, and the (scalar-valued) ESF takes the form of $\frac{d}{dx} \log p_t^{(n)}(x) = (\hat{x}_t^{(n)}(x) - x)/t$, where we define

$$\hat{x}_t^{(n)}(x) := \frac{p_{\mathcal{N}}(x-1; \sqrt{t}) - p_{\mathcal{N}}(x+1; \sqrt{t})}{p_{\mathcal{N}}(x-1; \sqrt{t}) + p_{\mathcal{N}}(x+1; \sqrt{t})} \tag{6}$$

While prior works have pointed out various smoothing effects arising from NN learning (Xu et al., 2019; Rahaman et al., 2019; Tirer et al., 2022), their training dynamics is highly difficult to analyze. In this work, our goal is less ambitious: we aim to show it is *possible* to cause the output of the diffusion model to interpolate the training data by smoothing the ESF. Thus, instead of dealing with NNs directly, we will consider a simple type of function smoothing as a proxy. In one dimension, given a function $f$ on $\mathbb{R}$ and a scalar $\tau > 0$, we define a $\tau$-smoothed version of $f$ to be a new function:

$$(\tau * f)(x) = \frac{1}{2\tau} \int_{x-\tau}^{x+\tau} f(x')dx' , \tag{7}$$

obtained by averaging $f$ over $[x - \tau, x + \tau]$. Interestingly, in Section 4, we will present empirical evidence that this simple form of function smoothing shares similarities with NN-learned estimators.

### 3.1 PIECE-WISE LINEAR APPROXIMATION AT SMALL $t$

The nonlinear denominator in (6) makes it hard to analyze the smoothing of the ESF exactly. Nonetheless, when $t$ is small, the noise variance is small and $\hat{x}_t^{(n)}(x)$ is close to $\text{sgn}(x)$. Thus, we will consider a piece-wise linear approximation of the ESF by $\bar{s}_t^{(n)}(x) = (\text{sgn}(x) - x)/t$. It is not hard to see that, $\forall \tau \in (0, 1)$,

$$(\tau * \bar{s}_t^{(n)})(x) = \hat{s}_{t,\tau}^{(n)}(x) , \tag{8}$$

where we further define

$$\hat{s}_{t,\tau}^{(n)}(x) := \begin{cases} -(x+1)/t , & \text{if } x \leq -\tau , \\ -(x-1)/t , & \text{if } x \geq \tau , \\ (1-\tau)x/(\tau t) , & \text{if } x \in [-\tau, \tau] . \end{cases} \tag{9}$$

One can in fact show that when $t$ is small, smoothed versions $\bar{s}_t^{(n)}$ and $\frac{d}{dx}\log p_t^{(n)}$ are close to each other. Quantitatively, the empirical $L^2$ approximation error (similar as in the training loss (4)) is exponentially small in $1/t$ as $t \to 0$:

**Lemma 1.** $\forall \tau \in (0,1)$, $\exists c, C > 0$ such that $\forall t \in (0,c)$, it holds that $t\mathbb{E}_{x \sim p_t^{(n)}}[\|\hat{s}_{t,\tau}^{(n)}(x) - (\tau *$
$\frac{d}{dx}\log p_t^{(n)})(x)\|^2] \leq C\exp(-\frac{(1-\tau)^2}{16t})$.

This lemma is proved in Appendix A.1. Hence, to understand the effect of smoothing the ESF, we will study smoothed versions of $\bar{s}_t^{(n)}$ as a proxy. As illustrated in Figure 1 (left), $\hat{s}_{t,\tau}^{(n)}$ is also piece-wise linear and $\hat{s}_{t,\tau}^{(n)} \equiv \bar{s}_t^{(n)}$ except on $[-\tau, \tau]$. Unlike $\bar{s}_t^{(n)}$, the maximum value of $|\hat{s}_{t,\tau}^{(n)}|$ on $[-1, 1]$ is $(1-\tau)/t < 1/t$ and is attained at $\pm\tau$. Thus, as $\tau$ increases from 0 to 1, the function $\hat{s}_{t,\tau}^{(n)}$ becomes smoother – e.g. as measured by its derivative's total variation, which equals $2/(\tau t)$ – while it deviates more and more from $\bar{s}_t^{(n)}$. Actually, as $t \to 0$, the empirical distribution $p_t^{(n)}$ becomes increasingly concentrated near $\pm 1$, and hence the difference between the two functions on $[-\tau, \tau]$ contributes less and less to their empirical $L^2$ discrepancy. In fact, the following result implies that as $t \to 0$, we can choose $\tau \to 1$ with $1 - \tau$ decreasing in proportion to $\sqrt{t}$ so that the approximation error as measured by the empirical $L^2$ discrepancy still remains on a constant order:

**Proposition 2.** There is a function $F : \mathbb{R} \to [0,1]$ such that, if $\tau_t \to 1$ as $t \to 0$ with $(1-\tau_t)^2/t = \kappa > 0$, then $\exists c, C > 0$ such that $\forall t \in (0,c)$,

$$\left| t\mathbb{E}_{x \sim p_t^{(n)}}\left[ \left|\hat{s}_{t,\tau_t}^{(n)}(x) - \tfrac{d}{dx}\log p_t^{(n)}(x)\right|^2 \right] - \tfrac{1}{2}F(\kappa) \right| \leq C\sqrt{t}. \tag{10}$$

Moreover, $F(\kappa)$ strictly decreases from 1 to 0 as $\kappa$ increases from 0 to $\infty$.

In other words, smoothing the ESF with a time-dependent scale $\tau_t = 1 - \sqrt{\kappa t}$ incurs a loss of (4) that is balanced across small $t$ asymptotically. This proposition is proved in Appendix A.2.

## 3.2 EFFECT ON THE DENOISING DYNAMICS

In light of Proposition 2, we will focus on a particular modification to the denoising dynamics where we replace the ESF by $\hat{s}_{t,\tau_t}^{(n)}$, with $\tau_t = 1 - \sqrt{\kappa t}$ for some $\kappa > 0$:

$$\frac{d}{dt}\mathbf{x}_t = -\tfrac{1}{2}\hat{s}_{t,\tau_t}^{(n)}(\mathbf{x}_t). \tag{11}$$

Thanks to the piece-wise linearity of (9), the backward-in-time dynamics of (11) can be solved analytically in terms of flow maps:

**Proposition 3.** For $t \in (0, 1/\kappa]$, the backward-in-time solution to (11) on $[0,t]$ is given by $\mathbf{x}_s = \phi_{s|t}(\mathbf{x}_t)$ for $0 \leq s \leq t$, where

$$\phi_{s|t}(x) = \begin{cases} \frac{\tau_s}{\tau_t}x, & \text{if } x \in [-\tau_t, \tau_t] \\ \sqrt{\frac{s}{t}}(x+1) - 1, & \text{if } x \leq -\tau_t \\ \sqrt{\frac{s}{t}}(x-1) + 1, & \text{if } x \geq \tau_t \end{cases} \tag{12}$$

The proposition is proved in Appendix A.5, and we illustrate the trajectories characterized by $\phi_{s|t}$ in Figure 2. The piece-wise linear nature of $\hat{s}_{t,\tau_t}^{(n)}$ divides the $x - \sqrt{t}$ plane into three regions (**A**, **B** and **C**) with linear boundaries, each defined by $x \leq -\tau_t$, $x \geq \tau_t$ and $-\tau_t \leq x \leq \tau_t$, respectively. Importantly, trajectories given by $\phi_{s|t}$ do not cross the region boundaries. If at $t_0 > 0$, $\mathbf{x}_{t_0}$ falls into region **A** (or **B**), then as $t$ decreases to 0, it will follow a linear path in the $x - \sqrt{t}$ plane to $y_1 = -1$ (or $y_2 = 1$). Meanwhile, if $\mathbf{x}_{t_0}$ falls into region **C**, then it will follow a linear path to the $x$-axis with a terminal value between $-1$ and 1. To determine this terminal value, we set $s = 0$ and obtain that

$$\phi_{0|t}(x) = \begin{cases} x/\tau_t, & \text{if } x \in [-\tau_t, \tau_t] \\ \text{sgn}(x), & \text{otherwise} \end{cases}$$

Notably, $\phi_{0|t}$ is invertible when restricted to $[-\tau_t, \tau_t]$. As a consequence, letting $\hat{p}_0^{(n)}$ and $\hat{p}_t^{(n)}$ denote the marginal distributions of $\mathbf{x}_0$ and $\mathbf{x}_t$, we derive that $\hat{p}_0^{(n)} = w_+\delta_1 + w_-\delta_{-1} + \hat{p}_{0;t}^{(n)}$, where

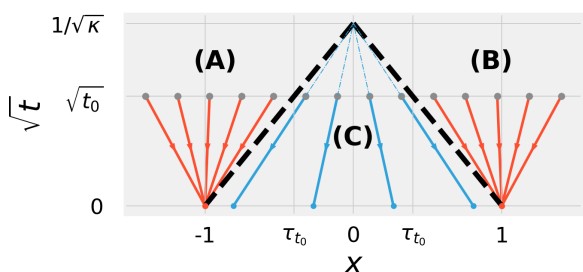

Figure 2: Illustration of the dynamics (11) under the smoothed score function studied in Section 3.2.

$w_+ := \mathbb{E}_{\mathbf{x} \sim p_t}[\mathbb{1}_{\mathbf{x} \geq \tau_t}]$, $w_- := \mathbb{E}_{\mathbf{x} \sim p_t}[\mathbb{1}_{\mathbf{x} \leq -\tau_t}]$, and

$$\hat{p}_{0;t}^{(n)}(x) := \begin{cases} \tau_t \hat{p}_t^{(n)}(\tau_t x), & \text{if } x \in [-1, 1] \\ 0, & \text{otherwise} \end{cases} \tag{13}$$

If $\hat{p}_t^{(n)}$ has positive density on $[-\tau_t, \tau_t]$, then $\hat{p}_{0;t}^{(n)}$ also has positive density $[-1, 1]$, corresponding to a smooth interpolation between the two training data points. Further, (13) also allows us to prove KL-divergence bounds for $\hat{p}_0^{(n)}$ based on those of $\hat{p}_t^{(n)}$, such as:

**Lemma 4.** $KL(u_{[-1,1]}||\hat{p}_0^{(n)}) = KL(u_{[-\tau_t, \tau_t]}||\hat{p}_t^{(n)})$, where $u_{[a,b]}$ means the uniform density on $[a, b]$.

As an example, we may consider the backward-in-time dynamics of (11) initialized with the marginal distribution $\hat{p}_{t_0}^{(n)} = p_{t_0}^{(n)}$ for some $t_0 \in (0, 1/\kappa)$. This corresponds to first evolving according to the *empirical* denoising dynamics down to $t_0$, and then in the rest of the dynamics replace the ESF with a smoothed approximation, $\hat{s}_{t,\tau_t}^{(n)} \approx (\tau_t * \frac{d}{dx} \log p_t^{(n)})$. In this case, we can leverage Lemma 4 to show that $\hat{p}_0^{(n)}$ has a finite KL-divergence from the uniform distribution on $[-1, 1]$:

**Corollary 5.** Suppose $\kappa > 0$ and $0 < t_0 < 1/\kappa$. If $\mathbf{x}_t$ solves (11) backward-in-time with $\mathbf{x}_{t_0} \sim p_{t_0}^{(n)}$, then there is $KL(u_{[-1,1]}||\hat{p}_0^{(n)}) \leq \frac{1}{3t_0(1-\sqrt{\kappa t_0})} + \log\left(\frac{\sqrt{t_0}}{1-\sqrt{\kappa t_0}}\right) + \log(2\sqrt{2\pi}) < \infty$.

In contrast, running the empirical denoising dynamics to time 0 results in the empirical distribution $p_0^{(n)}$, which is fully singular and has infinite KL-divergence with any smooth density on $[-1, 1]$.

### 3.3 GENERALIZATION TO MORE THAN TWO POINTS

The analysis above can be generalized to the scenario where $S$ consists of $n > 2$ points spaced uniformly on an interval $[-D, D]$, that is, $y_k := 2(k-1)\Delta - D$ for $k \in [n]$, where $\Delta := D/(n-1) = (y_{k+1} - y_k)/2$. We additionally define $a_k := y_k + \Delta = (y_k + y_{k+1})/2$ for $k \in [n-1]$. In this case, we replace (6) in the definition of the empirical score function by

$$\hat{x}_t^{(n)}(x) := \frac{\sum_{k=1}^n y_k p_{\mathcal{N}}(x - y_k, \sqrt{t})}{\sum_{k=1}^n p_{\mathcal{N}}(x - y_k, \sqrt{t})}, \tag{14}$$

and the piece-wise linear approximation to it at small $t$ is now given by

$$\bar{s}_t^{(n)}(x) := \begin{cases} (y_1 - x)/t, & \text{if } x \leq a_1, \\ (y_k - x)/t, & \text{if } x \in [a_{k-1}, a_k] \text{ for } k \in \{2, ..., n-1\}, \\ (y_n - x)/t, & \text{if } x \geq a_{n-1}. \end{cases} \tag{15}$$

Moreover, smoothed versions of $\bar{s}_t^{(n)}$ can still be written via (8) for $\tau \in (0, \Delta)$, where we now define

$$\hat{s}_{t,\tau}^{(n)}(x) := \begin{cases} (y_1 - x)/t, & \text{if } x \leq a_1 - \tau, \\ (y_n - x)/t, & \text{if } x \geq a_{n-1} + \tau, \\ (y_k - x)/t, & \text{if } x \in [a_{k-1} + \tau, a_k - \tau], \exists k \in [n], \\ (\Delta - \tau)(x - a_k)/(\tau t), & \text{if } x \in [a_k - \tau, a_k + \tau], \exists k \in [n-1], \end{cases} \tag{16}$$

and it is not hard to show that Proposition 2 continues to hold under modifications (details in Appendix A.2). Furthermore, the backward-in-time dynamics of (11) can also be solved analytically in a similar fashion, where (12) is replaced by $\phi_{s|t}(x) :=$

$$
\begin{cases}
\sqrt{\frac{s}{t}}(x - y_1) + y_1 , & \text{if } x \le y_1 + \sqrt{\kappa t} , \\
\sqrt{\frac{s}{t}}(x - y_n) + y_n , & \text{if } x \le y_n - \sqrt{\kappa t} , \\
\sqrt{\frac{s}{t}}(x - y_k) + y_k , & \text{if } x \in [y_k - \sqrt{\kappa t}, y_k + \sqrt{\kappa t}], \exists k \in \{2, ..., n-1\} , \\
\frac{\Delta - \sqrt{\kappa s}}{\Delta - \sqrt{\kappa t}}(x - a_k) + a_k , & \text{if } x \in [y_k + \sqrt{\kappa t}, y_{k+1} - \sqrt{\kappa t}], \exists k \in [n-1] .
\end{cases}
\tag{17}
$$

Setting $s = 0$, we see that

$$
\phi_{0|t}(x) = \begin{cases}
(\Delta x - a_k \sqrt{\kappa t})/(\Delta - \sqrt{\kappa t}) , & \text{if } x \in [y_k + \sqrt{\kappa t}, y_{k+1} - \sqrt{\kappa t}], \exists k \in [n-1] , \\
y_{\arg\min_k |y_k - x|} , & \text{otherwise.}
\end{cases}
$$

In particular, $\phi_{0|t}(x)$ is invertible when restricted to $\cup_{k \in [n-1]}[y_k + \sqrt{\kappa t}, y_{k+1} - \sqrt{\kappa t}]$. Hence, similar to the $n = 2$ case discussed above, if $p_t$ has positive density on $[-D, D]$, then so does $p_0$.

### 3.4 HIGHER DIMENSIONS: LINE SEGMENT AS HIDDEN SUBSPACE

Among different possible ways to generalize the notion of local smoothing introduced in Section 3 to higher dimensions, here we choose a simple one that averages over centered cubes. Namely, given $\tau > 0$ and a (vector-valued) function $\boldsymbol{f}$ on $\mathbb{R}^d$, we define for $\boldsymbol{x} = [x_1, ..., x_k]$ that

$$
(\tau * \boldsymbol{f})(\boldsymbol{x}) = \frac{1}{(2\tau)^d} \int_{x_1-\tau}^{x_1+\tau} ... \int_{x_d-\tau}^{x_d+\tau} \boldsymbol{f}(\boldsymbol{x}') dx_1' ... dx_d' .
\tag{18}
$$

Let us consider a case where $S = \{\boldsymbol{y}_k\}_{k \in [n]}$ consists of points that are spaced uniformly on the first axis of $\mathbb{R}^d$, that is, $\boldsymbol{y}_k = [y_{k,1}, 0, ..., 0]$, with $-D = y_{1,1} < ... < y_{n,1} = D$ and $\Delta$ defined in a similar way as in Section 3.3. In this case, for $\boldsymbol{x} = [x_1, ..., x_d]$, the noised empirical density is

$$
p_t^{(n)}(\boldsymbol{x}) = \left( \frac{1}{n} \sum_{k=1}^n p_{\mathcal{N}}(x_1 - y_{k,1}; \sqrt{t}) \right) \prod_{i=2}^d p_{\mathcal{N}}(x_i; \sqrt{t}) ,
$$

and the (vector-valued) ESF is given by $\nabla \log p_t^{(n)}(\boldsymbol{x}) = [\partial_1 \log p_t^{(n)}(\boldsymbol{x}), ..., \partial_d \log p_t^{(n)}(\boldsymbol{x})]$, where

$$
\partial_1 \log p_t^{(n)}(\boldsymbol{x}) = (\hat{x}_t^{(n)}(x_1) - x_1)/t ,
$$
$$
\forall i \in \{2, ..., n\} , \quad \partial_i \log p_t^{(n)}(\boldsymbol{x}) = -x_i/t ,
\tag{19}
$$

where $\hat{x}_t^{(n)}$ is defined in the same way as in (14) except for replacing each $y_k$ by $y_{k,1}$.

To study the smoothed ESF, $\tau * \nabla \log p_t^{(n)}$, we first observe that for each $i \in [d]$, $\partial_i \log p_t^{(n)}(\boldsymbol{x})$ depends only on $x_i$, and hence the repeated integral in (18) reduces to only the one in the $i$th dimension. For $i > 1$, $\partial_i \log p_t^{(n)}(\boldsymbol{x})$ is a linear function of $x_i$, which is invariant when averaged over a centered interval. For $i = 1$, $\partial_1 \log p_t^{(n)}(\boldsymbol{x})$ has the same form as in the $d = 1$ case where everything is projected onto the first dimension, and its smoothing can be expressed as $[\tau * \nabla \log p_t^{(n)}(\boldsymbol{x})]_1 = \frac{1}{2\tau t} \int_{x_1-\tau}^{x_1+\tau} (\hat{x}_t^{(n)}(x') - x') dx'$, which can be further approximated by $\hat{s}_{t,\tau}^{(n)}(x_1)$ on the same theoretical grounds as Lemma 1 and Proposition 2. Hence, in summary, we consider the following vector-valued function as a proxy for $\tau * \nabla \log p_t^{(n)}$:

$$
\hat{\boldsymbol{s}}_{t,\tau}^{(n)}(\boldsymbol{x}) := [\hat{s}_{t,\tau}^{(n)}(x_1), -x_2/t, ..., -x_d/t]^\mathsf{T} .
\tag{20}
$$

Similar to in Section 3.3, we consider a scenario where the smoothing occurs at a time-dependent scale, $\tau_t = \Delta - \sqrt{\kappa t}$. Thus, the dynamics can be approximately written as $\frac{d}{dt}\mathbf{x}_t = -\frac{1}{2}\hat{\boldsymbol{s}}_{t,\tau_t}^{(n)}(\boldsymbol{x})$, which is nicely decoupled across dimensions:

$$
\frac{d}{dt}\mathbf{x}_{t,1} = -\frac{1}{2}\hat{s}_{t,\tau_t}^{(n)}(\mathbf{x}_{t,1}) ,
\tag{21}
$$
$$
\forall i \in \{2, ..., n\} , \quad \frac{d}{dt}\mathbf{x}_{t,i} = \frac{1}{2}x_i/t .
\tag{22}
$$

Together, the $d$-dimensional system can be solved as follows:

**Proposition 6.** *For $t \in (0, 1/\kappa]$, the backward-in-time solution to (21, 22) on $[0, t]$ is given by $\mathbf{x}_s = \Phi_{s|t}(\mathbf{x}_t) = [\phi_{s|t}(\mathbf{x}_{t,1}), 0, ..., 0]$ for $0 \leq s \leq t$, where $\phi_{s|t}$ is defined in the same way as in (12).*

In summary, we see distinct dynamical behaviors in the first and the rest of the dimensions. As $t \to 0$, all except for the first dimension shrinks to zero at a rate of $\sqrt{t}$, corresponding to a uniform collapse of the $d$-dimensional space onto the $x_1$-axis. Meanwhile, the dynamics in the first dimension is qualitatively similar to the $d = 1$ case. In particular, if the marginal distribution of $\mathbf{x}_{t,1}$ has a positive density on $[-D, D]$, then so will $\mathbf{x}_{0,1}$, which implies that $\mathbf{x}_0$ admits a non-singular density that interpolates smoothly among the training data on the desired one-dimensional subspace.

**On early stopping.** This behavior is different from what can be achieved by denoising under the exact ESF, either by running it fully to $t = 0$ or by stopping it at some positive $t_{\min}$ (i.e., early stopping): the former leads to the collapse onto the training data points (i.e., full memorization), while in the latter case the terminal distribution is still spread over all $d$ dimensions and equivalent to adding Gaussian noise to the training data points. In either case, the terminal distribution has infinite KL-divergence from any smooth density supported on the one-dimensional subspace.

We note that the result above does not yet constitute manifold recovery in a stronger sense, because the definition (18) depends on choice of the coordinate system while the analysis only applies when the hidden subspace is one of the axes. To achieve generality, we need to consider alternative definitions of smoothing in higher dimensions that is invariant to coordinate rotations. We discuss one such example in the Appendix A.3, which achieves a similar effect on the ESF and results in the same dynamics as (21) and (22).

## 4 NUMERICAL ILLUSTRATIONS

### 4.1 SCORE FUNCTIONS LEARNED BY NNS

To examine the smoothing effect of NN learning empirically, we compare NN-learned score functions with those obtained by local smoothing in the setting of Section 3 with $d = 1$ and $n = 2$. For the former, we train three-layer MLPs to fit the ESF under regularization on the model parameters with various strengths, and additional details are given in Appendix A.7. As shown in Figure 1 (center and right), the score estimators learned by NNs are nearly piece-wise linear and can be matched closely by our $\hat{s}_{t,\tau}$ under suitable choices of the smoothing scale, which increases along with the strength of regularization. This provides empirical evidence that our theoretical analyses based on locally smoothing the score function may indeed be relevant to understanding NN-based diffusion models.

### 4.2 INTERPOLATION AS A DYNAMICAL EFFECT OF SCORE SMOOTHING

To illustrate the connection between the interpolation effect and score smoothing, we perform numerical experiments under the setup studied in Section 3.4 where training data are spaced uniformly on the first axis. We choose $d = 2$, $n = 4$ and $D = 1$, and compare the outcomes of the denoising dynamics (2) under four different choices for the score function: **(i)** the exact ESF ($s_t = \nabla \log p_t^{(n)}$), **(ii)** the $\tau_t$-smoothed ESF ($s_t = \tau_t * \nabla \log p_t^{(n)}$), **(iii)** the piece-wise linear approximation to the $\tau_t$-smoothed ESF ($s_t = \hat{s}_{t,\tau_t}^{(n)}$ from (20)), and **(iv)** an NN-learned score function ($s_t = s_t^{\mathrm{NN}}$).

All four denoising processes are initialized at $t_0 = 0.02$ with the same marginal distribution $\mathbf{x}_{t_0} \sim p_{t_0}^{(n)}$, the noised empirical distribution at $t_0$, which means **(i)** is equivalent to the empirical denoising dynamics. As in Section 3.4, we set the smoothing scale in **(ii)** and **(iii)** to be $\tau_t = \Delta - \sqrt{\kappa t}$ with $\kappa = 0.4$. The ESF is computed from its analytical expression (14), and smoothing is computed numerically through a Monte-Carlo approximation to the integral (18) with 512 samples. To ensure numerical stability at small $t$, we truncate the sampled values of $\nabla p_t^{(n)}$ based on magnitude. At $t_0$, 4096 realizations of $\mathbf{x}_{t_0}$ are sampled from $p_{t_0}^{(n)}$. Then, the ODEs are solved (backward-in-time) numerically using Euler's method with step size 0.002 after a change of variable $t \to \sqrt{t}$. For **(iv)**, we parameterize $s_t^{\mathrm{NN}}(\boldsymbol{x})$ by a three-layer MLP applied to both $t$ and $\boldsymbol{x}$ and train it to minimize a discretized version of (4) *without* regularization. Details on its training can be found in Appendix A.7.

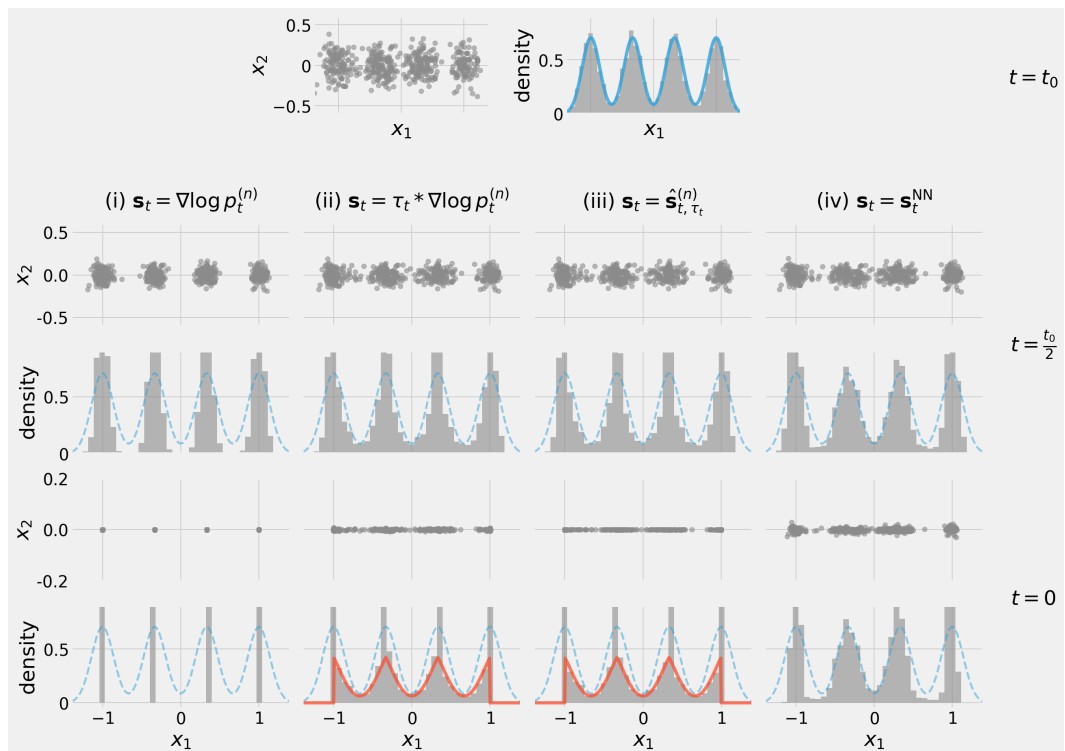

Figure 3: Illustrations of the numerical experiments discussed in Section 4.2 with $d = 2$ and $n = 4$ via snapshots at different $t$. Each column shows the denoising process under one of four choices of the score function: **(i)** the exact ESF; **(ii)** the $\tau_t$-smoothed ESF; **(iii)** the piece-wise linear approximation to the $\tau_t$-smoothed ESF; **(iv)** an NN-learned score function. The denoising processes all start from the marginal distribution $p_{t_0}^{(n)}$ at $t = t_0$ and evolve backward-in-time to $t = 0$ according to the respective score function. At each of $t = t_0$, $t_0/2$ and $0$, we plot the samples from the denoising processes in $\mathbb{R}^2$ and the density histograms of the samples projected onto the first dimension. In the latter, the blue and the red curves are the analytical predictions of $p_{t_0}^{(n)}$ and the non-singular part of $\hat{p}_0^{(n)}$ (through an extension of (13) to the case where $n = 4$), respectively.

**Results.** The results are illustrated in Figure 3. At $t = t_0$, in all cases, the samples are distributed as $p_{t_0}^{(n)}$, a mixture of isotropic Gaussian distributions centered at each of the training data points. As the denoising dynamics progresses, across **(i)**−**(iii)**, the variance along the second dimension shrinks gradually to zero (slightly positive in **(ii)** due to numerical errors) at a rate that is roughly identical, consistent with the fact that all three share the same underlying ODE for the second dimension, (22). Meanwhile, in contrast with the empirical denoising process of **(i)**, where the variance along the first dimension shrinks to zero as well, we see in **(ii)** and **(iii)** that the variance along the first dimension remains positive for all $t$, validating the interpolation effect of smoothing the ESF. Remarkably, the behaviors of **(ii)** and **(iii)** remain close throughout the dynamics, providing empirical support of the piece-wise linear approximation at small $t$ in addition to Lemma 1. Moreover, at $t = 0$, the density histograms of **(ii)** and **(iii)** are well-matched by our analytical prediction of the non-singular component of $\hat{p}_0^{(n)}$ in Section 3, providing further validation of our theoretical analysis. Lastly, we observe a similar interpolation effect in **(iv)** as in **(ii)** and **(iii)** where the generated samples are less concentrated on the training set but converge to near zero in the second dimension.

## 5 DISCUSSIONS AND LIMITATIONS

The main motivation of the work is not to propose new designs for diffusion models (an example in this regard is Scarvelis et al. 2023) but to understand *why* existing ones can produce new samples different from the training set. We focus on one hypothesis, that the ability comes from a smoothing

of the score function when learned by NNs, and have provided mathematical and empirical evidence that score smoothing is indeed *sufficient* to induce such an ability through an interpolation effect on the outcome of the denoising dynamics. At the core of this phenomenon are two properties of the smoothed ESF: reducing the speed of convergence towards training data along the *tangential* direction (to avoid memorization) while preserving it along the *normal* direction (to ensure a convergence onto the subspace). The decomposition of the score function into tangential and normal components when data belong to subspaces was also studied by Chen et al. (2023b); Wang & Vastola (2023b). To our knowledge, our work is novel in showing that the distinct effect of score smoothing on the two components leads to data interpolation along the subspace.

**Limitations.** To further validate of this hypothesis, another important question is whether (and how) score smoothing is achieved in practice, which is not the focus of the current work. There has been empirical evidence that NNs indeed tend to learn smoother versions of the ESF (Aithal et al., 2024), though theoretical understandings are still lacking and, admittedly, there is no justification as to why the smoothing effect caused by NN training should be similar to the type of local smoothing studied in this paper. We nevertheless hope that our analysis can help to elucidate the phenomenon and provide intuitions for future work along this line. Another limitation of the present work is the vastly simplified setup compared to real-world scenarios, which enables us to obtain insights through mathematically solvable models. It would be very interesting to extend our theory to cases where training data belong to multi-dimensional and nonlinear manifolds as well as to more general types of diffusion models (De Bortoli et al., 2021; Albergo et al., 2023; Lipman et al., 2023; Liu et al., 2023).

## 6 RELATED WORKS

**Generalization vs memorization in diffusion models.** Several works have studied empirically the generalization vs memorization behaviors in diffusion generative models, observing the transition from the former to the latter when the model capacity increases relatively to the training set size (Gu et al., 2023; Yi et al., 2023; Carlini et al., 2023; Kadkhodaie et al., 2024). Li et al. (2024) showed that learning the ESF well does not result in generalization, which stresses the discrepancy between it and the score functions of the *true* underlying density. When provided with an oracle estimator for the latter, it has indeed been proved that diffusion models can produce accurate distributions (Song et al., 2021a; Lee et al., 2022; De Bortoli, 2022; Chen et al., 2023a;c; Benton et al., 2024; Huang et al., 2024), but the question remains as to how to estimate the true score functions well. When assuming that the ground truth density or its score function belongs to certain restricted classes (e.g, densities being Gaussian mixtures, or sub-Gaussian, or belonging to certain smoothness classes; their score functions being Lipschitz or belonging to reproducing kernel Hilbert spaces), prior works have constructed score estimators with guaranteed sample complexity and sometimes statistical optimality (Block et al., 2020; Li et al., 2023; Zhang et al., 2024; Wibisono et al., 2024; Chen et al., 2024; Gatmiry et al., 2024). Due to their assumptions, these results do not typically apply to the scenario considered in Section 3.4 where the data are supported on low-dimensional sub-manifolds. A more fundamental distinction is that, whereas these results concern the estimation of densities from i.i.d. samples, our work does not assume underlying densities and make the finite training set a starting point of analysis. Both perspectives have advantages: the former setting is suitable for deriving sample complexity bounds (but needs assumptions on the underlying distribution), whereas the latter allows us to exploit the local geometry and derive analytical solutions of the denoising dynamics. In future works, it will be interesting to also extend our analysis to the density estimation perspective.

**Diffusion models and the manifold hypothesis.** An influential hypothesis in machine learning is that high-dimensional real-world data often lie in low-dimensional sub-manifolds (Tenenbaum et al., 2000; Fefferman et al., 2016), and it has been argued that diffusion models are able to estimate their intrinsic dimensions (Stanczuk et al., 2024) or learn manifold features in order of descending variance (Wang & Vastola, 2023a;b). Assuming conditions on the score estimator, Pidstrigach (2022); De Bortoli (2022) studied the convergence of diffusion models in the case of manifold data. Oko et al. (2023); Chen et al. (2023b) proved sample complexity guarantees for score estimation using certain NN models under the manifold hypothesis. Nonetheless, the error bound by Oko et al. (2023) is in Wasserstein distance and hence less informative about the memorization phenomenon than KL divergence (e.g., unlike Corollary 5, a finite Wasserstein distance bound does not exclude the distribution produced by the model from being fully singular). Meanwhile, Chen et al. (2023b)

focused on a regime where the denoising dynamics is stopped at a fixed $t_{\min} > 0$ while $n \to \infty$, and hence unlike Corollary 5, their result does not guarantee the recovery of a density restricted to the subspace with finite training data. The intricate interplay between $n$ and $t$ and their effect on memorization have been nicely studied by Biroli et al. (2024) in the case of Gaussian mixture data.

**Score smoothing.** Aithal et al. (2024) showed empirically that NNs tend to learn smoother versions of the ESF and argued that this leads to a mode interpolation effect that underlies model hallucination. Scarvelis et al. (2023) designed alternative closed-form diffusion models by smoothing the ESF, although the theoretical analysis therein is limited to showing that their smoothed score function is directed towards certain barycenters of the training data. Inspired by their work, in this paper we further analyze mathematically the effect of score smoothing on the denoising dynamics.

In summary, our work shows mathematically how smoothing the ESF leads the denoising dynamics to produce distributions that interpolate among the training data on their subspace, elucidating a mechanism of potential relevance to both the generalization ability and the hallucination phenomenon of score-based diffusion models.

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

# A  APPENDIX

**Notations**   For functions $f, g : \mathbb{R}_+ \to \mathbb{R}_+$, we will write $f(t) = O(g(t))$ if $\exists c, C > 0$ such that $\forall t \in (0, c)$, it holds that $f(t) \le Cg(t)$.

## A.1  GENERALIZATION OF LEMMA 1 AND ITS PROOF

**Lemma 7.** *Under the setting considered in Section 3.3 with $S = \{y_k\}_{k \in [n]} \subseteq [-D, D]$, $\forall \tau \in [0, \Delta]$, there is $t\mathbb{E}_{x \sim p_t^{(n)}}[\|(\tau * \bar{s}_t^{(n)})(x) - (\tau * \frac{d}{dx} \log p_t^{(n)})(x)\|^2] = O(\exp(-\frac{(\Delta - \tau)^2}{16t}))$.*

*Proof*: By the definition of $p_t^{(n)}$, it suffices to show that $\forall k \in [n]$,

$$\int |(\tau * \tfrac{d}{dx} \log p_t^{(n)})(x) - (\tau * \bar{s}_t^{(n)})(x)|^2 p_\mathcal{N}(x - y_k; \sqrt{t})dx = O(\exp(-(\Delta - \tau)^2/(16t))) \, .$$

Consider any $k \in [n]$. The left-hand-side above can be rewritten as

$$\int_{-\infty}^{\infty} \left| \frac{1}{2\tau} \int_{x-\tau}^{x+\tau} \tfrac{d}{dx} \log p_t^{(n)}(x')dx' - \frac{1}{2\tau} \int_{x-\tau}^{x+\tau} \bar{s}_t^{(n)}(x')dx' \right|^2 p_\mathcal{N}(x - y_k; \sqrt{t})dx$$

$$= \int_{-\infty}^{\infty} \left| \frac{1}{2\tau} \int_{x-\tau}^{x+\tau} \left( \tfrac{d}{dx} \log p_t^{(n)}(x') - \bar{s}_t^{(n)}(x') \right) dx' \right|^2 p_\mathcal{N}(x - y_k; \sqrt{t})dx$$

$$\le \int_{-\infty}^{\infty} \frac{1}{2\tau} \int_{x-\tau}^{x+\tau} \left| \tfrac{d}{dx} \log p_t^{(n)}(x') - \bar{s}_t^{(n)}(x') \right|^2 dx' p_\mathcal{N}(x - y_k; \sqrt{t})dx$$

$$= \int_{-\infty}^{\infty} \left| \tfrac{d}{dx} \log p_t^{(n)}(x') - \bar{s}_t^{(n)}(x') \right|^2 \left( \frac{1}{2\tau} \int_{x'-\tau}^{x'+\tau} p_\mathcal{N}(x' - y_k; \sqrt{t})dx \right) dx'$$

If $x \ge y_k + \frac{1}{2}(\Delta + \tau) > y_k + \tau$, there is $\sup_{x \in [x'-\tau, x'+\tau]} p_\mathcal{N}(x - y_k; \sqrt{t}) \le p_\mathcal{N}(x' - y_k - \tau; \sqrt{t})$. Hence, also noticing that $|\frac{d}{dx} \log p_t^{(n)}(x)|, |\bar{s}_t^{(n)}(x)| \le (|x| + 2D)/t$, we obtain that

$$\int_{y_k + \frac{1}{2}(\Delta + \tau)}^{\infty} \left| \tfrac{d}{dx} \log p_t^{(n)}(x') - \bar{s}_t^{(n)}(x') \right|^2 \left( \frac{1}{2\tau} \int_{x'-\tau}^{x'+\tau} p_\mathcal{N}(x - y_k; \sqrt{t})dx \right) dx'$$

$$\le \frac{1}{\sqrt{2\pi t}} \int_{y_k + \frac{1}{2}(\Delta + \tau)}^{\infty} \left| \tfrac{d}{dx} \log p_t^{(n)}(x') - \bar{s}_t^{(n)}(x') \right|^2 \exp\left( -\frac{(x' - y_k - \tau)^2}{2t} \right) dx$$

$$\le \frac{1}{\sqrt{2\pi t}} \int_{y_k + \frac{1}{2}(\Delta + \tau)}^{\infty} \frac{4(|x'| + 2D)^2}{t^2} \exp\left( -\frac{(x' - y_k - \tau)^2}{2t} \right) dx$$

$$\le \frac{1}{\sqrt{2\pi t}} \int_{y_k + \frac{1}{2}(\Delta + \tau)}^{\infty} \frac{4(|x' - y_k - \tau| + 4D)^2}{t^2} \exp\left( -\frac{(x' - y_k - \tau)^2}{2t} \right) dx$$

$$\le \frac{8}{\sqrt{2\pi t^3}} \int_{y_k + \frac{1}{2}(\Delta + \tau)}^{\infty} \left( \left( \frac{x' - y_k - \tau}{\sqrt{t}} \right)^2 + 16D^2 \right) \exp\left( -\frac{(x' - y_k - \tau)^2}{2t} \right) dx \tag{23}$$

$$= \frac{8}{\sqrt{2\pi t^3}} \int_{\frac{\Delta - \tau}{2\sqrt{t}}}^{\infty} (x^2 + 16D^2) \exp\left( -x^2/2 \right) dx$$

$$\le \frac{8}{\sqrt{2\pi t^3}} \left( 16D^2 \sqrt{\frac{\pi}{2}} + \frac{\Delta - \tau}{2\sqrt{t}} \right) \exp\left( -\frac{(\Delta - \tau)^2}{8t} \right)$$

$$= O\left( t^{-2} \exp\left( -\frac{(\Delta - \tau)^2}{8t} \right) \right)$$

A similar bound can be derived when the outer integral is integrated from $-\infty$ to $y_k - \frac{1}{2}(\Delta + \tau)$.

Next, suppose $x \in [y_k - \frac{1}{2}(\Delta + \tau), y_k + \frac{1}{2}(\Delta + \tau)]$. Then there is $|x - y_k| \leq \frac{1}{2}(\Delta + \tau)$ while $|x - y_k| \geq \frac{3}{2}\Delta - \frac{1}{2}\tau$ for $l \neq k$. Thus, it holds for any $l \neq k$ that

$$\frac{p_{\mathcal{N}}(x - y_k; \sqrt{t})}{p_{\mathcal{N}}(x - x_l; \sqrt{t})} = \exp\left(-\frac{|x - y_k|^2 - |x - x_l^2|}{2t}\right)$$

$$\geq \exp\left(\frac{\Delta(\Delta - \tau)}{t}\right)$$

Hence, writing $q_{t,k}(x) := \frac{p_{\mathcal{N}}(x - y_k; \sqrt{t})}{\sum_{l=1}^{n} p_{\mathcal{N}}(x - x_l; \sqrt{t})}$, there is $q_{t,k}(x) \geq 1 - (n-1)\exp\left(-\frac{\Delta(\Delta - \tau)}{t}\right)$ and for $l \neq k$, $q_{t,l}(x) < \exp\left(-\frac{\Delta(\Delta - \tau)}{t}\right)$. Therefore,

$$\left| s_t^{(n)}(x) - \bar{s}_{t,\tau}^{(n)}(x) \right| \leq \frac{|(q_{t,k}(x) - 1)y_k| + \sum_{l \neq k} |q_{t,k}(x)y_k|}{t}$$

$$\leq \frac{2(n-1)D}{t} \exp\left(-\frac{\Delta(\Delta - \tau)}{t}\right) = O\left(t^{-1}\exp\left(-\frac{\Delta(\Delta - \tau)}{t}\right)\right). \tag{24}$$

Since $\frac{1}{2\tau}\int_{x'-\tau}^{x'+\tau} p_{\mathcal{N}}(x - y_k; \sqrt{t})dx \leq \frac{1}{2\pi}$ for any $x'$, we then have

$$\int_{y_k - \frac{1}{2}(\Delta+\tau)}^{y_k + \frac{1}{2}(\Delta+\tau)} \left|\frac{d}{dx} \log p_t^{(n)}(x') - \bar{s}_t^{(n)}(x')\right|^2 \left(\frac{1}{2\tau}\int_{x'-\tau}^{x'+\tau} p_{\mathcal{N}}(x - y_k; \sqrt{t})dx\right) dx'$$

$$\leq \frac{\Delta + \tau}{2\tau} \sup_{y_k - \frac{1}{2}(\Delta+\tau) \leq x' \leq y_k + \frac{1}{2}(\Delta+\tau)} \left|s_\sigma^{(n)}(x) - \bar{s}_{\sigma,k}^{(n)}(x)\right|^2 \tag{25}$$

$$= O\left(t^{-2}\exp\left(-\frac{2\Delta(\Delta - \tau)}{t}\right)\right)$$

Combining (23) with (25) yields the desired result.

$\square$

## A.2 Generalization of Proposition 2 and its proof

**Proposition 8.** *There is a function $F : \mathbb{R} \to [0, 1]$ such that, if $t \to 0$ and $\tau \to \Delta$ with $(\Delta - \tau)^2/t = \kappa > 0$, then there is*

$$t\mathbb{E}_{x \sim p_t^{(n)}}\left[|\hat{s}_{t,\tau}^{(n)}(x) - \frac{d}{dx}\log p_t^{(n)}(x)|^2\right] = \frac{n-1}{n}F(\kappa) + O(\sqrt{t}). \tag{26}$$

*Moreover, $F(\kappa)$ strictly decreases from 1 to 0 as $\kappa$ increases from 0 to $\infty$.*

*Proof*: We will write $\delta = \Delta - \tau$ and $\sigma = \sqrt{t}$ for simplicity. By Lemma 1, we only need to show that

$$t\int |\hat{s}_{t,\tau}^{(n)}(x) - \bar{s}_t^{(n)}(x)|^2 p_\sigma^{(n)}(x)dx = \frac{n-1}{n}F(\kappa) + O(\sqrt{t}).$$

By the definition of $p_\sigma^{(n)}$, it suffices to separately consider the integral with respect to the density $p_{\mathcal{N}}(x - x_k; \sigma)$ for each $k \in [n]$. We define

$$x_{k,-} = \begin{cases} -\infty, & \text{if } k = 1 \\ x_k - \delta, & \text{otherwise} \end{cases}, \qquad x_{k,+} = \begin{cases} \infty, & \text{if } k = n \\ x_k + \delta, & \text{otherwise} \end{cases}$$

By construction, $\hat{s}_{t,\tau}^{(n)}$ is a piece-wise linear function where the slope is changed only at each $x_{k,-}$ and $x_{k,+}$. In particular, for $k \in [n]$, there is $\hat{s}_{t,\tau}^{(n)}(x) = \bar{s}_t^{(n)}(x)$ when $x \in [x_{k,-}, x_{k,+}]$. Hence, we only need to estimate the difference between the two outside of $[x_{k,-}, x_{k,+}]$.

We first consider the interval $[x_{k,+}, x_k + \Delta] = [x_k + \delta, x_k + \Delta]$, on which it holds that

$$\hat{s}_{t,\tau}^{(n)}(x) - \bar{s}_t^{(n)}(x) = \frac{\Delta}{t} \cdot \frac{x - (x_k + \delta)}{\tau}, \tag{27}$$

by the linearity of the two functions. Hence,

$$
t \int_{x_k+\delta}^{x_k+\Delta} |\hat{s}_{t,\tau}^{(n)}(x) - \bar{s}_t^{(n)}(x)|^2 p_{\mathcal{N}}(x - x_k; \sigma) dx
$$

$$
= \left(\frac{\Delta}{\Delta - \delta}\right)^2 \int_{x_k+\delta}^{x_k+\Delta} \left|\frac{x - x_k}{\sigma} - \frac{\delta}{\sigma}\right|^2 p_{\mathcal{N}}(x - x_k; \sigma) dx
$$

$$
= \left(\frac{\Delta}{\Delta - \delta}\right)^2 \left(\int_{x_k+\delta}^{\infty} \left|\frac{x - x_k}{\sigma} - \frac{\delta}{\sigma}\right|^2 p_{\mathcal{N}}(x - x_k; \sigma) dx \right.
$$

$$
\left. - \int_{x_k+\Delta}^{\infty} \left|\frac{x - x_k}{\sigma} - \frac{\delta}{\sigma}\right|^2 p_{\mathcal{N}}(x - x_k; \sigma) dx\right) \tag{28}
$$

Note that

$$
\int_{x_k+\delta}^{\infty} \left|\frac{x - x_k}{\sigma} - \frac{\delta}{\sigma}\right|^2 p_{\mathcal{N}}(x - x_k; \sigma) dx = \frac{1}{2} F(\kappa) , \tag{29}
$$

where we define $F(\kappa) := 2 \int_{\kappa}^{\infty} |u - \kappa|^2 p_{\mathcal{N}}(u; 1) du$. It is straightforward to see that, as $\kappa$ increases from 0 to $\infty$, $F$ strictly decreases from 1 to 0. Therefore,

$$
t \int_{x_k+\kappa\sigma}^{x_k+\Delta} |\hat{s}_{t,\tau}^{(n)}(x) - \bar{s}_t^{(n)}(x)|^2 p_{\mathcal{N}}(x - x_k; \sigma) dx
$$

$$
= \left(\frac{\Delta}{\tau}\right)^2 \left(F(\kappa) - \int_{\Delta/\sigma}^{\infty} |u - \kappa|^2 p_{\mathcal{N}}(u; 1) dx\right) \tag{30}
$$

$$
= \frac{1}{2} F(\kappa) + O(\sigma)
$$

Next, we consider the interval $[x_k + \Delta, \infty)$, in which we have

$$
|\hat{s}_{t,\tau}^{(n)}(x) - \bar{s}_t^{(n)}(x)| \leq \frac{\Delta}{t} . \tag{31}
$$

Thus,

$$
t \int_{x_k+\Delta}^{\infty} |\hat{s}_{t,\tau}^{(n)}(x) - \bar{s}_t^{(n)}(x)|^2 p_{\mathcal{N}}(x - x_k; \sigma) dx \leq t \int_{x_k+\Delta}^{\infty} \left|\frac{\Delta}{t}\right|^2 p_{\mathcal{N}}(x - x_k; \sigma) dx
$$

$$
= \frac{\Delta^2}{t} \int_{\Delta/\delta}^{\infty} p_{\mathcal{N}}(u; 1) du \tag{32}
$$

$$
= O\left(\frac{\Delta^2}{t} \exp\left(-\frac{\Delta^2}{2t}\right)\right) .
$$

Hence, we have

$$
t \int_{x_k+\kappa\delta}^{\infty} |\hat{s}_{t,\tau}^{(n)}(x) - \bar{s}_t^{(n)}(x)|^2 p_{\mathcal{N}}(x - x_k; \sigma) dx = \frac{1}{2} F(\kappa) + O(\sigma) . \tag{33}
$$

Similarly, for $k \in \{2, ..., n\}$, we can show that

$$
t \int_{-\infty}^{x_k-\kappa\delta} |\hat{s}_{t,\tau}^{(n)}(x) - \bar{s}_t^{(n)}(x)|^2 p_{\mathcal{N}}(x - x_k; \sigma) dx = \frac{1}{2} F(\kappa) + O(\sigma) . \tag{34}
$$

Thus, there is

$$
t \int_{-\infty}^{\infty} |\hat{s}_{t,\tau}^{(n)}(x) - \bar{s}_t^{(n)}(x)|^2 p_{\mathcal{N}}(x - x_k; \sigma) dx = \begin{cases} F(\kappa) + O(\sigma) , & \text{if } k \in \{2, ..., n-1\} \\ \frac{1}{2} F(\kappa) + O(\sigma) , & \text{if } k = 1 \text{ or } n . \end{cases} \tag{35}
$$

Summing them together, we get that

$$
t \int_{-\infty}^{\infty} |\hat{s}_{t,\tau}^{(n)}(x) - \bar{s}_t^{(n)}(x)|^2 p_{\sigma}^{(n)}(x) dx = \frac{n-1}{n} F(\kappa) + O(\sigma) . \tag{36}
$$

## A.3 ALTERNATIVE DEFINITION OF LOCAL SMOOTHING IN HIGHER DIMENSIONS

Given a compact set of vectors, $A \subseteq \mathbb{R}^d$, we define its *element with minimum Euclidean norm* as

$$\gamma(A) := \begin{cases} \arg\min_{\boldsymbol{v} \in A} \|\boldsymbol{v}\|_2 , & \text{if } \arg\min_{\boldsymbol{v} \in A} \|\boldsymbol{v}\|_2 \text{ is unique} \\ \boldsymbol{0} , & \text{otherwise} \end{cases}$$

Given $\tau > 0$ and a vector-valued function $\boldsymbol{f}$ on $\mathbb{R}^d$, as an alternative to (18), we define

$$(\tau * \boldsymbol{f})(\boldsymbol{x}) := \gamma \left( \left\{ \frac{1}{2\tau} \int_{-\tau}^{\tau} \boldsymbol{f}(\boldsymbol{x} + r\boldsymbol{v}) dr : \boldsymbol{v} \in \mathbb{S}^{d-1} \right\} \right) . \tag{37}$$

Note that this definition is invariant to translations and rotations to the Euclidean space.

As in the main text, we will approximate the first component of the ESF, $\partial_1 \log p_t^{(n)}$, by a piece-wise linear function when $t$ is small, therefore focusing on:

$$\bar{\boldsymbol{s}}_t^{(n)}(\boldsymbol{x}) := [\bar{s}_t^{(n)}(x_1), -x_2/t, ..., -x_d/t]^\mathsf{T} \approx \nabla \log p_t^{(n)}(\boldsymbol{x}) ,$$

as a proxy to the ESF.

To ease notations, we write $\boldsymbol{s}_{\tau,\boldsymbol{v}}(\boldsymbol{x}) := \frac{1}{2\tau} \int_{-\tau}^{\tau} \bar{\boldsymbol{s}}_t^{(n)}(\boldsymbol{x} + r\boldsymbol{v}) dr$. For $i > 1$, thanks to the linearity of $\partial_i \log p_t^{(n)}(\boldsymbol{x})$, we have that $[\boldsymbol{s}_{\tau,\boldsymbol{v}}(\boldsymbol{x})]_i = \partial_i \log p_t^{(n)}(\boldsymbol{x}) = -x_i/t$. For $i = 1$, there is

$$[\boldsymbol{s}_{\tau,\boldsymbol{v}}(\boldsymbol{x})]_1 = \frac{1}{2\tau t} \int_{-\tau}^{\tau} \bar{s}_t^{(n)}(x_1 + rv_1) dr$$

$$= \frac{1}{2\tau |v_1| t} \int_{-\tau|v_1|}^{\tau|v_1|} \bar{s}_t^{(n)}(x_1 + \tilde{r}) d\tilde{r}$$

$$= ((\tau|v_1|) * \bar{s}_t^{(n)})(x_1) .$$

Therefore, we obtain that

$$\left\{ \boldsymbol{s}_{\tau,\boldsymbol{v}}(\boldsymbol{x}) : \boldsymbol{v} \in \mathbb{S}^{d-1} \right\} = \left\{ [\hat{s}_{t,\tilde{\tau}}^{(n)}(x_1), -x_2/t, ..., -x_d/t]^\mathsf{T} : \tilde{\tau} \in [0, \tau] \right\} .$$

From (9) and Figure 1, it is clear that for any fixed $x$, $\hat{s}_{t,\tilde{\tau}}^{(n)}(x)$ keeps the same sign as $\tilde{\tau}$ ranges from 0 to $\tau$ while its absolute value decreases. Therefore, we derive that

$$(\tau * \bar{\boldsymbol{s}}_t^{(n)})(\boldsymbol{x}) = \gamma(\{ \boldsymbol{s}_{\tau,\boldsymbol{v}}(\boldsymbol{x}) : \boldsymbol{v} \in \mathbb{S}^{d-1} \}) = [\hat{s}_{t,\tau}^{(n)}(x_1), -x_2/t, ..., -x_d/t]^\mathsf{T} .$$

Hence, if we choose $\tau_t = \Delta - \sqrt{\kappa t}$, we see that the dynamics described by (21) and (22) indeed agrees with

$$\frac{d}{dt} \mathbf{x}_t = -\frac{1}{2} (\tau * \bar{\boldsymbol{s}}_t^{(n)})(\mathbf{x}_t) ,$$

under the alternative definition of (37) as well.

## A.4 PROOF OF LEMMA 4

$$\mathrm{KL}(u_{[-1,1]} \| p_0) = \int_{-1}^{1} \frac{1}{2} \cdot (-\log 2 - \log(p_0(x))) dx$$

$$= -\log 2 - \frac{1}{2} \int_{-1}^{1} \log(\tilde{p}_t(x)) dx$$

$$= -\log 2 - \frac{1}{2(1 - \sqrt{\kappa t})} \int_{-1+\sqrt{\kappa t}}^{1-\sqrt{\kappa t}} \left( \log(1 - \sqrt{\kappa t}) + \log(p_t(x')) \right) dx'$$

$$= \frac{1}{2(1 - \sqrt{\kappa t})} \int_{-1+\sqrt{\kappa t}}^{1-\sqrt{\kappa t}} \log(1/(2(1 - \sqrt{\kappa t}))) - \log(p_t(x')) dx'$$

$$= \mathrm{KL}(u_{[-1+\sqrt{\kappa t}, 1-\sqrt{\kappa t}]} \| p_t)$$

$\square$

### A.5 PROOF OF PROPOSITION 3

We consider each of three cases separately.

**Case I: $x \in [-1 + \sqrt{\kappa t}, 1 - \sqrt{\kappa t}]$.** In this case, it is easy to verify that $x_s = \frac{1 - \sqrt{\kappa s}}{1 - \sqrt{\kappa t}} x$ is a valid solution to the ODE

$$\frac{d}{ds} x_s = -\frac{1}{2} \frac{\sqrt{\kappa s}}{1 - \sqrt{\kappa s}} \frac{x_s}{s} \,,$$

on $[0, t]$ that satisfies the terminal condition $x_t = x$. It remains to verify that for all $s \in (0, t)$, it holds that $x_s \in [-1 + \sqrt{\kappa s}, 1 - \sqrt{\kappa s}]$ (i.e., the entire trajectory during $[0, t]$ remains in region B).

Suppose that $x \geq 0$, in which case it is clear that $x_s \geq 0, \forall s \in [0, t]$. Moreover, it holds that

$$x_s - (1 - \sqrt{\kappa s}) = \frac{1 - \sqrt{\kappa s}}{1 - \sqrt{\kappa t}} (x_t - (1 - \sqrt{\kappa t})) \leq 0$$

Therefore, $x_s \in [0, 1 - \sqrt{\kappa s}] \subseteq [-1 + \sqrt{\kappa s}, 1 - \sqrt{\kappa s}]$. A similar argument can be made if $x < 0$.

**Case II: $x \leq -1 + \sqrt{\kappa t}$.** In this case, it is also easy to verify that $x_s = \sqrt{\frac{s}{t}}(x + 1) - 1$ is a valid solution to the ODE

$$\frac{d}{ds} x_s = \frac{1}{2} \frac{x + 1}{s} \,,$$

on $[0, t]$ that satisfies the terminal condition $x_t = x$. It remains to verify that for all $s \in (0, t)$, it holds that $x_s \leq -1 + \sqrt{\kappa s}$ (i.e., the entire trajectory during $[0, t]$ remains in region A). This is obvious because

$$(x_s + 1) - \sqrt{\kappa s} = \sqrt{\frac{s}{t}}(x + 1) - \sqrt{\kappa s} = \sqrt{\frac{s}{t}}(x + 1 - \sqrt{\kappa t}) \leq 0 \,.$$

**Case III: $x \geq 1 - \sqrt{\kappa t}$.** A similar argument can be made as in Case II above.

$\square$

### A.6 PROOF OF COROLLARY 5

By symmetry, we only need to consider the right half of the interval, $[0, \tau_{t_0}]$, on which there is $p_{t_0}(x) = p_{t_0}^{(n)}(x) \geq \frac{1}{2} p_\mathcal{N}(x - 1; \sqrt{t_0})$. There is

$$\int_0^{\tau_{t_0}} \log \left( p_\mathcal{N}(x - 1; \sqrt{t_0}) \right) dx = \int_{-1}^{-\sqrt{\kappa t_0}} \log \left( \frac{1}{\sqrt{2\pi t_0}} \exp(-x^2/t_0) \right) dx$$

$$= -\frac{1 - \sqrt{\kappa t_0}}{2} (\log(2\pi) + \log(t_0)) - \frac{1}{t_0} \int_{-1}^{-\sqrt{\kappa t}} x^2 dx$$

$$\geq -\frac{1 - \sqrt{\kappa t_0}}{2} (\log(2\pi) + \log(t_0)) - \frac{1}{3 t_0} \,.$$

Therefore,

$$\mathrm{KL}(u_{[0, \tau_{t_0}]} || p_{t_0}) = \frac{1}{\tau_{t_0}} \int_0^{\tau_{t_0}} \log(1/\tau_{t_0}) - \log \left( \frac{1}{2} p_\mathcal{N}(x - 1; \sqrt{t_0}) \right) dx$$

$$\leq -\log(1 - \sqrt{\kappa t}) + \log(2) - \frac{1}{1 - \sqrt{\kappa t_0}} \left( -\frac{1 - \sqrt{\kappa t_0}}{2} (\log(2\pi) + \log(t_0)) - \frac{1}{3 t_0} \right)$$

$$\leq \frac{1}{3 t_0 (1 - \sqrt{\kappa t_0})} + \log \left( \frac{\sqrt{t_0}}{1 - \sqrt{\kappa t_0}} \right) + \log(2\sqrt{2\pi}) \,.$$

By symmetry, the same bound can be obtained for $\mathrm{KL}(u_{[-\tau_{t_0}, \tau_{t_0}]} || p_{t_0})$, which yields the desired result when combined with Lemma 4.

## A.7 ADDITIONAL DETAILS ON THE NUMERICAL EXPERIMENTS

**Figure 1 (left)**: We chose $t = 0.1$ and $\tau = 0.58$.

**Figure 1 (center)**: We trained three-layer MLP to fit the ESF at $t = 0.1$. The model is trained by the Adam optimizer with learning rate $0.001$ for $1200$ steps and using weight decay (i.e., $L^2$ regularization on the model parameters) with various strengths, $\lambda$. At each training step, the optimization objective is an approximation of the expectation that appears in 4 based on a batch of $1024$ samples from $p_t^{(n)}$. We considered three choices of $\lambda$: $\lambda_1 = 0.005$, $\lambda_2 = 0.01$ and $\lambda_3 = 0.02$.

**Figure 1 (right)**: We chose $t = 0.1$ and three values of $\tau$: $\tau_1 = 0.35$, $\tau_2 = 0.42$, and $\tau_3 = 0.53$, which were picked manually to match the corresponding curves in Figure 1 (center).

**Figure 3**: After rescaling it by $\sqrt{t}$, we parameterize the score function by a three-layer MLP applied to the concatenation of $\log(t)$ and $\boldsymbol{x}$, where the log transform serves to reduce the effect of weight sharing $t$ when $t$ is close to zero. We train the model to fit the ESF for $t \in [0, t_0]$ with $t_0 = 0.02$. At each step, the optimization objective is an approximation of the integral in 4 (with $T$ set to be $t_0$) based on sampling $t$ uniformly from $[0, t_0]$ and then $\boldsymbol{x}$ from $p_t^{(n)}$. We used the Adam optimizer with learning rate $0.004$, batch size $16$ and a total number of $3000$ steps.

