# OpenReview forum: "On the Interpolation Effect of Score Smoothing"
_ICLR.cc/2025/Conference — ICLR 2025 Conference Withdrawn Submission_

### Official Review · Reviewer_be3W · 2024-10-29

**Soundness:** 3
**Presentation:** 3
**Contribution:** 3
**Rating:** 6
**Confidence:** 3

**Summary:**

This paper investigates the interpolation effect that arises when smoothing the empirical score function (ESF) in score-based diffusion models, specifically exploring its impact on generative sample diversity. The authors focus on a theoretical model where training data lie in a one-dimensional subspace and use a local smoothing approach on the ESF, demonstrating that smoothing allows for interpolation between training points and mitigates memorization. They offer a mathematical analysis under simplified settings and provide numerical illustrations to validate their findings. Their method seeks to clarify why score smoothing may improve generalization in generative models by facilitating smoother transitions within the data's underlying structure.

**Strengths:**

The paper presents a novel perspective on how score smoothing enables generative models to interpolate between data points, avoiding full memorization—a valuable insight for understanding generalization in diffusion models. Numerical experiments illustrate the model’s interpolation behavior, validating that a smoothed ESF leads to sample distributions that interpolate among training points. This supports the theoretical analysis and enhances understanding of the smoothing effects on generative dynamics.

**Weaknesses:**

1. A typo at line 57: $t$ should be inside $\sqrt{\cdot}$.

2. The motivation for the chosen local smoothing technique is not entirely intuitive. While they study an approximation in the $L^2$ sense, it remains unclear why this specific smoothing function is preferable over alternative smoothing methods. The paper acknowledges this limitation but could further clarify. See questions below.

3. The relationship between $\kappa$ and $t$ in section 3 is a bit confusing. See questions below.

**Questions:**

1. The motivation for the chosen local smoothing technique is not entirely intuitive. The author shows that their smoothed score function is close to the true score in the $L^2$ sense, which is usually assumed for the NNs in practice. If one can further show that 'if two different score functions are both close to the true score in the $L^2$ sense, then the dynamics driven by them are close to each other', then studying a specific smoothed score could directly help us understanding the others.

2. The relationship between $\kappa$ and $t$ in section 3 is a bit confusing. In Proposition 2, $\kappa$ is defined by $t$, while in proposition 3, $t$ is restricted in $[0, \kappa]$. In practice, should $t$ be specified first?

---

### Official Review · Reviewer_hCgf · 2024-11-03

**Soundness:** 3
**Presentation:** 3
**Contribution:** 3
**Rating:** 6
**Confidence:** 4

**Summary:**

The paper investigated the smoothing effect in score estimation and presented theoretical analyses on a simplified data model. Numerical results were also presented to support the argument.

**Strengths:**

The paper is well-organized and has a clear presentation. The theoretical results on two points and one-dimensional subspace are intuitive and easy to follow.

**Weaknesses:**

It is not clear how the analyses of one one-dimensional subspace can be extended to more practical data.
- For example, how can the analyses be connected back to address memorization/hallucination behaviors mentioned in the introduction?
- In line 475, I cannot agree the analysis is "distribution-agnostic," as it still relies on a highly simplified assumption of the practical data.

**Questions:**

More broadly, the smoothing effect can be related to generalization error and the inductive bias of deep neural networks. Could the authors briefly comment on how the analyses can help understand these phenomena?

---

### Official Review · Reviewer_mS12 · 2024-11-04

**Soundness:** 1
**Presentation:** 2
**Contribution:** 1
**Rating:** 5
**Confidence:** 4

**Summary:**

This work studies the the interpolation effect of score smoothing and provides numerical experiments.

**Strengths:**

The paper considers a simple one-dimensional model to study the smoothed score function. Mathematical properties are derived and numerical experiments are conducted.

**Weaknesses:**

Major questions:
1. The biggest concern is the gap between the problem that the authors attempt to study and the model proposed in this paper. The starting point is memorization vs. generalization in diffusion models. The authors conjecture the memorization is from the smoothed score function. It raised two natural questions: 1) how do you know the NN-learning score estimator has such smoothing effect in practice? 2) Even though the score estimator used in practice is smoothed, how is this property linked to the generalization, i.e., overcoming the memorization? These two questions are not well-studied in the paper.
2. The paper considers a(n) (essentially) one-dimensional model. In this case, the (empirical) score function has a closed-form. However, the authors do not study this score function while turning to its smoothed version. Then I would like to ask, even if we know all the properties of the smoothed score function, how can we have more knowledge about the true score function, even in this simple case?
3. The authors conduct numerical experiments to compare three score functions/estimators. However, it is not clear how to interpret the results. The distribution after adding noise must be the same as standard Gaussian is the stationary distribution in all three cases. Then, what is the new information by plotting the density and histograms?

Minor questions:
1. Why do you introduce $\hat{x}_t^{(n)}$, $\bar{s}_t^{(n)}$ and $$\widehat{s}^{(n)}_{t, \tau}$$, three versions of the score function? Also, if the target distribution is $p_0^{(n)}$, why to you need a smoothed version $\hat{p}_t^{(n)}$? Please clarify the relation between them and emphasize the necessity of introducing these auxiliary functions/distributions.
2. The target distribution studies seems to be relevant to Gaussian mixture models. There are some papers studying learing GMMs using diffusion models, e.g., https://arxiv.org/abs/2404.18869 and https://arxiv.org/abs/2404.18893. Any relation to the literature?

**Questions:**

See weaknesses.

---

### Official Review · Reviewer_75YM · 2024-11-11

**Soundness:** 3
**Presentation:** 4
**Contribution:** 3
**Rating:** 6
**Confidence:** 3

**Summary:**

This paper theoretically explores  the score of variance exploding SDEs (VE-SDEs) to showcase the success in score based diffusion models by arguing that the smoothing of the data-score generating novel samples that interpolates across the training data subspace.

**Strengths:**

1. The paper starts of a simpler and more accessible 1d 2 data points analysis to motivate their insights to the reader before moving onto the more technical results.
2. The paper's insights into their smoothed ESF approximation seem valuable to the community. In particular, it shows that the KL regarding a uniform distribution is bounded (unlike if using the ESF directly) and quite neat.

**Weaknesses:**

I think the paper could do a better job at motivating their results for a wider audience:

1. Equation (8) can and should be defined before Lemma 1 , in Lemma 1 you throw in the smoothed score defined in an implicit way and then you redefine it and label it it equation 8, it would be better to introduce equation 8 first and then use the notation defined in equation 8 to introduce lemma 1.
2. It seems plausible that a network would learn a smoothed version of the score (as discussed in limitations), but you never strongly motivate / discuss this, earlier discussion of this would motivate your choice of analysis better.  Either way it would be helpful for you to motivate and introduce the smoothed score a bit earlier in the text.

**Questions:**

1. Line 057 $\sigma$ should be $t$ instead? otherwise there's no reference to $t$ in the abbreviation/RHS of the equation.
2. Whilst big O notation is standard, I think its a bit cleaner and more common to write these results  with $\leq$
3. As mentioned in the weakness its clear the smoothed score induces a controllable loss, but I think to complete the story the authors need to discuss / motivate why would we learn anything akin to the specific choice of smoothed score used in this work ?  I understand this is mentioned in the limitations but maybe something earlier in the introduction would be helpful.

---

### Note · Authors · 2025-09-19

I have read and agree with the venue's withdrawal policy on behalf of myself and my co-authors.

---

### Meta-Review · Area_Chair_fUx7 · 2024-12-18

**Metareview:**

This paper investigates the interpolation effect due to smoothing the score function in diffusion models. The authors consider a rather basic model where the training data is restricted to a 1d subspace, they perform a theoretical analysis (in which the empirical score function has a closed form) and then validate such analysis with numerical results.

The reviewers agree that the topic is interesting, the paper is clear and well written, and the results bring some interesting insights. However, such insights are hindered by the strong assumptions required by the theory (unidimensional data). The gap between the theory and the setting that we authors wish to analyze is quite significant and this constitutes the main weakness of the manuscript, as raised by reviewer mS12. I also think this is major weakness and therefore recommend a rejection at this stage.

I do think that the approach of the paper has potential and I would encourage the authors to pursue this line of work, providing a more general analysis and resubmitting an improved version to a future venue.

**Additional Comments On Reviewer Discussion:**

The response of the authors did mitigate some of the concerns raised by the reviewers, but not really the main issue pointed out by reviewer mS12 about the strong assumptions needed by the theoretical analysis.

---

### Decision · Program_Chairs · 2025-01-22

Reject